# Structure and inhibition of the human lysosomal transporter Sialin

Philip Schmiege [1], Linda Donnelly [1], Nadia Elghobashi-Meinhardt[2], Chia-Hsueh Lee [3] & Xiaochun Li [1,4] ✉

Sialin, a member of the solute carrier 17 (SLC17) transporter family, is unique in its ability to transport not only sialic acid using a pH-driven mechanism, but also transport mono and diacidic neurotransmitters, such as glutamate and N-acetylaspartylglutamate (NAAG), into synaptic vesicles via a membrane potential-driven mechanism. While most transporters utilize one of these mechanisms, the structural basis of how Sialin transports substrates using both remains unclear. Here, we present the cryogenic electron-microscopy structures of human Sialin: apo cytosol-open, apo lumen-open, NAAG–bound, and inhibitor–bound. Our structures show that a positively charged cytosol-open vestibule accommodates either NAAG or the Sialin inhibitor Fmoc-Leu-OH, while its luminal cavity potentially binds sialic acid. Moreover, functional analyses along with molecular dynamics simulations identify key residues in binding sialic acid and NAAG. Thus, our findings uncover the essential conformational states in NAAG and sialic acid transport, demonstrating a working model of SLC17 transporters.

Lysosomes play a crucial role in cellular waste management by breaking down and recycling both extracellular materials taken up by endocytosis and obsolete internal cellular components and aggregates[1,2]. The dysfunction of lysosomal transport proteins can contribute to rare genetic disorders known as lysosomal storage diseases (LSDs)[3]. Understanding the mechanisms of these key macromolecule transport systems will aid in future therapeutic development to treat their associated LSDs. Sialin is a unique membrane transporter in both the variety of substrates it transports and the mechanisms it uses for transport[4,5]. It was originally identified as the main transporter for importing free sialic acid from the lysosomal lumen to the cytosol after the degradation of glycoproteins and glycolipids in the lysosome[6]. Mutations in Sialin, which can lead to the accumulation of free sialic acid in the lysosomal lumen, are responsible for two devastating neurodegenerative sialic acid storage disorders: Salla Disease and Infantile Sialic acid Storage Disease (ISSD)[4,5].

Sialin (SLC17A5) belongs to the solute carrier 17 (SLC17) transporter subfamily that includes nine related transmembrane major facility superfamily (MFS) transporters, all of which are dedicated to anion transport, including the vesicular glutamate transporters 1-3 (VGLUT1–3) and a vesicular nucleotide transporter (VNUT)[7,8]. Sialin is the only protein in the SLC17 family that transports sialic acid and the only one that is ubiquitously expressed[9]. While Sialin was originally identified as a sialic acid transporter, several studies have shown that it in fact can transport a variety of other substrates, including nitrate[10], and the neurotransmitters aspartate, glutamate[11,12], and N-acetylaspartylglutamate (NAAG)[13]. In addition to being a key N-terminal glycan that is utilized in various cell processes, sialic acid is also abundant in the human brain and plays a crucial role in both neural transmission and the structure of gangliosides involved in synaptogenesis[14]. NAAG acts as a neuromodulator of glutamatergic synapses via triggering the activation of presynaptic metabotropic glutamate receptor[15]. An animal study showed that NAAG levels in the brain decreased in Sialin-deficient mice, suggesting that Sialin is a major vesicular transporter for NAAG[13]. Therefore, Sialin is regarded as an essential transporter in neuronal activity.

[1]Department of Molecular Genetics, University of Texas Southwestern Medical Center, Dallas, TX, USA. [2]School of Chemistry, University College Dublin, Belfield, Dublin, Ireland. [3]Department of Structural Biology, St. Jude Children's Research Hospital, Memphis, TN, USA. [4]Department of Biophysics, University of Texas Southwestern Medical Center, Dallas, TX, USA. ✉e-mail: xiaochun.li@utsouthwestern.edu

Sialin transports sialic acid and nitrate from the lysosomal lumen to cytosol using the pH gradient[6,10,11]. Previous studies have shown that sialic acid transport depends on proton-coupled electroneutral process with a 1:1 (sialic acid : H[+]) stoichiometry and this transport does not interfere with the membrane potential or the other inorganic ion gradients[5,16]. In contrast, previous assays showed that Sialin employs a membrane potential ($\Delta\psi$)-driven mechanism, rather than a pH gradient mechanism, for transporting neurotransmitters such as aspartate, glutamate, and NAAG from the cytosol into synaptic vesicles[12,13].

Although the structures of VGLUT2[17] and DgoT[18] (a bacterial homolog of SLC17 transporters) have been studied, and despite the apo cytosol-open structure of Sialin being reported recently[19], it remains unclear how Sialin recognizes its distinct substrates and how it undergoes its conformational changes. Here, we capture multiple states of human Sialin up to 2.8 Å resolution using cryo-electron microscopy (cryo-EM) to illuminate the transport cycle of Sialin in the lysosome and synaptic vesicles.

## Results
### Overall structures of Sialin
To determine the structure of Sialin, we expressed the wild-type human protein with an N-terminal FLAG-tag (Sialin[WT]) in HEK293 cells and purified the protein using anti-Flag M2 resin in the presence of detergent. Owing to the low molecular weight of Sialin, ~55 kDa, and lack of discernible features outside of the detergent micelle, initial

structural studies by cryo-EM did not yield high-resolution data. Therefore, we reconstituted purified protein into liposomes and injected them into mice to generate Sialin antibodies. Antibodies were screened for structural specificity by selecting ELISA-positive but denatured-negative samples[20]. We successfully identified a monoclonal antibody denoted 10B8 that specifically binds to native Sialin. We expressed and purified Fab[10B8] from HEK293 cells and subsequently assembled the Sialin-Fab[10B8] complex. The resulting complex migrates as a single peak on the gel filtration at both pH 5.0 and pH 7.5 (Supplementary Fig. 1).

The cryo-EM structures of Sialin[WT] in the apo state were determined at 3.2-Å (pH 5.0) and 2.8-Å (pH 7.5) resolution respectively (Fig. 1a, b, Supplementary Fig. 2 and Table 1). Sialin adopts a classical MFS fold consisting of 12 transmembrane helices (TMs) with two pseudo-symmetric domains: N-domain (TMs 1-6) and C-domain (TMs 7-12) (Fig. 1c). The Fab[10B8] binds to the cytosolic helix which links the two domains (Fig. 1a, b and Supplementary Fig. 3a). Unlike our previous finding on the pH-driven lysosomal cystine transporter cystinosin[21], the structures of Sialin[WT] in pH 5.0 and pH 7.5 reveal a similar cytosol-open conformation. The overall structure of Sialin[WT] has the same fold with the recently determined cytosol-open structure of human Sialin[19], with a root mean square deviation (R.M.S.D.) value of 0.5 Å (Supplementary Fig. 3b). A Dali search[22] shows that the overall structure of Sialin[WT] shares a similar fold with D-galactonate:proton symporter DgoT[18] and nitrate/nitrite antiporter NarK[23] with R.M.S.D. values of 2.4 Å and 2.7 Å (Supplementary Fig. 3c, d).

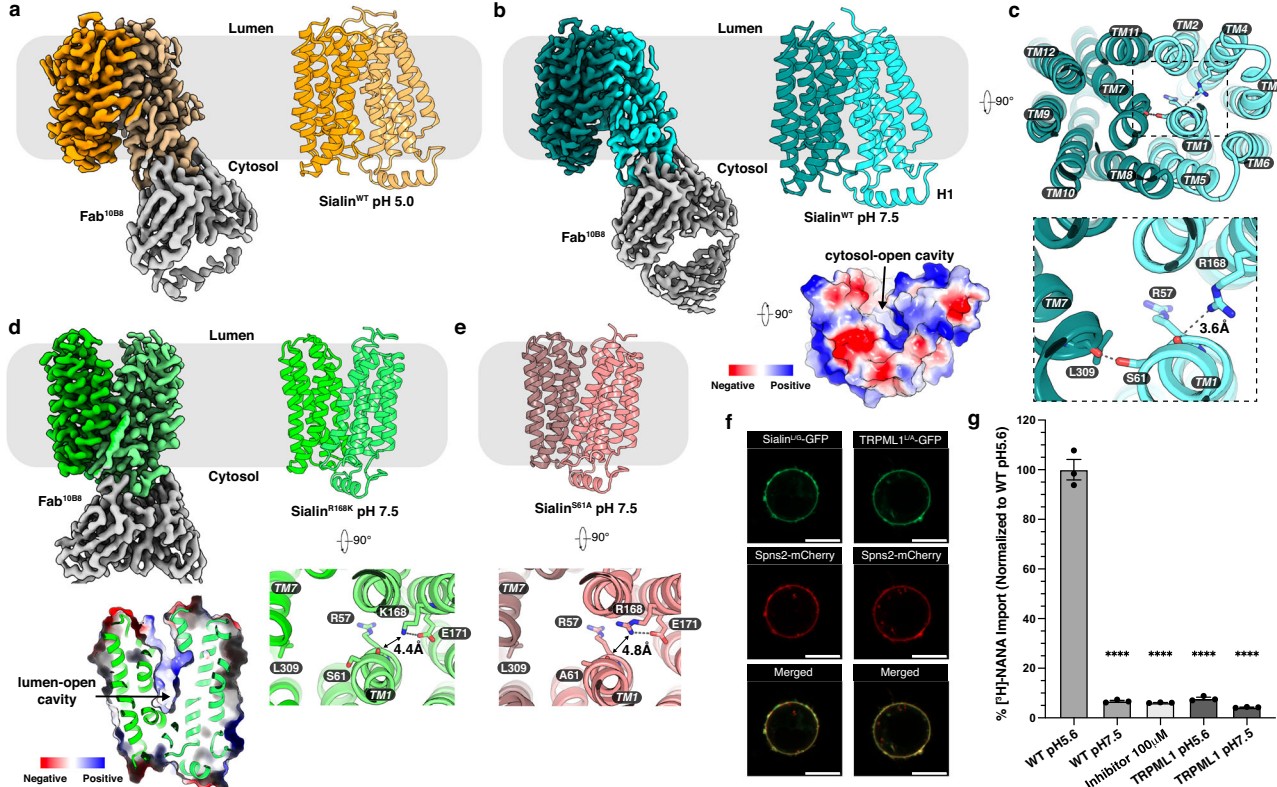

**Fig. 1 | Overall structures of human Sialin. a** Overall structure of Sialin[WT] in a cytosol-open state at pH 5.0 viewed from the side of the membrane. **b** Overall structure of Sialin[WT] in a cytosol-open state at pH 7.5 viewed from the side of the membrane (left) or from the cytosolic space (lower right). **c** The luminal view of Sialin[WT]. The interaction details between R57, S61, R168, and L309 are shown. TMs and related residues are labeled. **d** Overall structure of Sialin[R168K] in a lumen-open state. The surface representation of the lumen-open cavity is indicated (bottom left). **e** Overall structure of Sialin[S61A] in a lumen-open state. Residues R57, A61, R168, and L309 are shown from the luminal view (panel **d** and **e**). **f** Representative images

of the cellular localization of Sialin[GFP/LL]. Sialin[GFP/LL] or TRPML1[L/A] are co-overexpressed with Spns2-mCherry (plasma membrane control) in HEK293 (n = 3 biological replicates). Scale bar, 10 μm. **g** Functional validation of human Sialin[L/G] sialic acid transport. The wild type is defined as WT. Data are represented as mean ± SEM (n = 3 biological replicates). P values are all in comparison to WT pH 5.6 and are: 2.7e-11, 2.6e-11, 3.1e-11, and 2.1e-11 in order. An ordinary one-way ANOVA with Dunnett's multiple comparisons test was used to generate the statistics in GraphPad Prism 10. ****P ≤ 0.0001.

To capture the lumen-open state, we introduced a mutation on Arg168. This residue has a hydrophilic interaction with the carbonyl of Arg57, stabilizing TM1 which in turn presumably confines the conformation of TM7 in the cytosol-open state via a direct interaction between the hydroxyl group of Ser61 and the main chain carbonyl of Leu309 (Fig. 1c). The cryo-EM structure of the Sialin[R168K] complex with Fab[10B8] in the apo state at pH 7.5 was determined at 3.3-Å resolution and exhibits a lumen-open conformation (Fig. 1d, Supplementary Figs. 1c and 4, and Table 1). Lys168 forms a salt bridge with Glu171 and does not bind to the main chain carbonyl of Arg57 (Fig. 1d). Moreover, we determined the cryo-EM structure of the Sialin[S61A] in complex with Fab[10B8] at 3.2-Å resolution (Fig. 1e, Supplementary Figs. 1d and 4, and Table 1). The structure of Sialin[S61A] is in a lumen-open state. The interaction between Arg57 and Arg168 is disrupted, instead, Arg168 forms the salt bridge with Glu171, which is consistent with our observation on Sialin[R168K]. These findings support our hypothesis of the hydrophilic interaction network in the luminal leaflet to modulate the conformational change of Sialin. Dali search shows that the overall structure of Sialin[R168K] shares a similar fold with, VGLUT2 and monocarboxylate transporter 1[24] (MCT1) with R.M.S.D. values of 1.2 Å and 2.4 Å (Supplementary Fig. 3e, f).

Immunofluorescence studies showed late endosomal and lysosomal localization of Sialin in HEK293 cells; therefore, similar to previous research, we employed a surface-expressed Sialin variant (Sialin[L/G]) with the sorting motif mutations (L22G/L23G) and an N-terminal GFP tag to validate the function of Sialin by cell-based radiolabeled transport assays for both sialic acid transport[5] and NAAG transport (Supplementary Fig. 5a). The transport activity of each variant or condition was normalized by protein expression level, which was quantitatively measured by GFP fluorescence. Sialin[L/G] and its variants are expressed on the cell surface like the sphingosine-1-phosphate transporter Spns2[25] (Fig. 1f and Supplementary Fig. 5b). Our results show robust sialic acid transport activity in pH 5.6 but not pH 7.5, supporting previous conclusions that Sialin functions as a proton/sialic acid symporter[5] (Fig. 1g). The activity of Sialin can be mostly inhibited in the presence of the previously described Sialin inhibitor Fmoc-Leu-OH[26] at a final concentration of 100 μM. Therefore, we used the Sialin[L/G] as a wild-type control to validate the function of each Sialin variant in the sialic acid transport, and all the mutations in the functional assays were introduced into Sialin[L/G].

## Substrate translocation pathway

To achieve alternating access transport, MFS transporters usually oscillate between lumen (outward)-open and cytosol (inward)-open states[27]. The Sialin structures in both the lumen-open and cytosol-open conformations reveal a solvent-accessible cavity, which forms the substrate translocation pathway, at the center of the interface between the N-domain and C-domain (Fig. 1b and d). During the transport cycle, the interactions between two domains regulate the access of distinct substrates to the lipid bilayer[27]. Overlaying the two states shows that the N-domain of each structure aligns well, while the bulk of the movements are in the C-domain (Fig. 2a).

In the lumen-open state, the cytosolic end of TM8 shifts more than 12 Å towards the N-domain around the kink at Ser332 on the lumen leaflet (Fig. 2a), and the luminal end of TM7 and TM11 shifts by over 9-Å away from the N-domain (Fig. 2b). These movements result in a pronounced opening of the substrate translocation pathway to the cytosol. The lumen-open structure reveals that the hydrophilic interactions between Arg195 on TM5 and Gln273 on the cytosolic helix, Ser196 on TM5 and Arg353 on TM8, Lys197 on TM5 and Asp350 on TM8 seal the substrate translocation pathway from the cytosolic space (Fig. 2c). In the cytosol-open state, Phe116 on TM2 forms π-π interaction with Tyr306 on TM7 and Gln123 on TM2 interacts with Thr434 on TM11, leading to a collapse of the translocation pathway in the luminal leaflet

(Fig. 2d). The interaction between Tyr335 on TM8 and Leu309 on TM7 ensures that Leu309 is in a certain conformation to interact with Val58 and Ser61 on TM1 to further lock Sialin in the cytosol-open state (Fig. 2d). To functionally test these interactions, we mutated Ser61, Tyr306, Leu309 and Tyr335 to alanine individually. As a result, the sialic acid transport activity was considerably reduced, indicating that these residues play critical roles in the transport cycle (Fig. 2e). The further structural analysis showed that Sialin has two glutamate residues (Glu171 and Glu175) accessible to the solvent from the lumen (Fig. 2f), which may play a role in proton sensing.

## A putative sialic acid binding site

Previous studies showed that Sialin has a fairly low apparent affinity for sialic acid with a Km of about 1 mM[5,16]. To capture the sialic acid-bound state, we incubated Sialin[WT] and Sialin[R168K] with N-acetylneuraminic acid (NANA) at a final concentration of 5 mM before grid preparation. But the resulting cryo-EM map did not reveal an unambiguous density in the lumen-open cavity. Therefore, we employed Glide[28] to dock the sialic acid into the cavity. The top three positions are similar, suggesting that His298, Tyr301, and Ser411 form hydrophilic interactions with sialic acid (Fig. 3a). Compared to the structures of galactonate bound DgoT and a cyclic analog of glutamate (APCD) bound VGLUT2[29], all the substrates are engaged by the lumen-open cavities (Fig. 3b). Our functional assays show that mutating His298, Tyr301 or Ser411 to alanine can attenuate the transport activity, supporting the roles of these residues in sialic acid recognition (Fig. 3c and Supplementary Fig. 5). To validate our model, we conducted molecular dynamics (MD) simulations of NANA-bound Sialin. The results showed that over a 100 ns timescale, NANA moves within a few nanoseconds to a position that is stabilized by the interactions with His298, Tyr301, and Ser411 (Fig. 3d and Supplementary Movie 1), supporting our model.

## Fmoc-Leu-OH bound structure

Recently, a study showed that Fmoc-Leu-OH functions as a noncompetitive inhibitor of sialic acid transport in Sialin[26]. Interestingly, this study demonstrated that a derivative of Fmoc-Leu-OH (named compound 45) can rescue the trafficking defect of a disease-causing mutant Sialin[R39C] (Supplementary Fig. 6a), potentially contributing to the development of pharmacological molecules for treating Salla disease[26]. To reveal the binding mode of Fmoc-Leu-OH, we determined the structure of Fmoc-Leu-OH bound Sialin[WT] in the cytosol-open state at 3.67-Å resolution (Fig. 4a, Supplementary Fig. 7, and Table 1). Despite the relatively lower overall resolution, the local cryo-EM density of Fmoc-Leu-OH was clearly resolved in the cytosol-open cavity and matched its chemical structure, so we modeled it into the cryo-EM map (Fig. 4a, b).

Tyr54, Phe179, Tyr203, Tyr301, Phe305 and Tyr306 create a hydrophobic cavity to host the aromatic rings of Fmoc-Leu-OH (Fig. 4c and Supplementary Fig. 7e). The extensive hydrophobic contacts between the Sialin and Fmoc-Leu-OH lock Sialin in the cytosol-open state and prevent the access of substrates into the translocation pathway. We modeled compound 45 from the previous study into the structure based on the current Fmoc-Leu-OH position (Supplementary Fig. 6d). Our model shows that the Fmoc groups of Fmoc-Leu-OH and compound 45 can be aligned and the coumarinyl group of the compound 45 is accommodated by the cytosol-open cavity (Supplementary Fig. 6d). Our modeling result is different from the previous prediction[26], but determining the precise binding mode of this compound requires further investigation.

## NAAG bound structure

To dissect the mechanism of Sialin-mediated export of neurotransmitters, such as NAAG and glutamate, we purified Sialin[WT] in the presence of NAAG at 0.2 mM final concentration. Before grid preparation, we also supplemented NAAG into the protein at a final

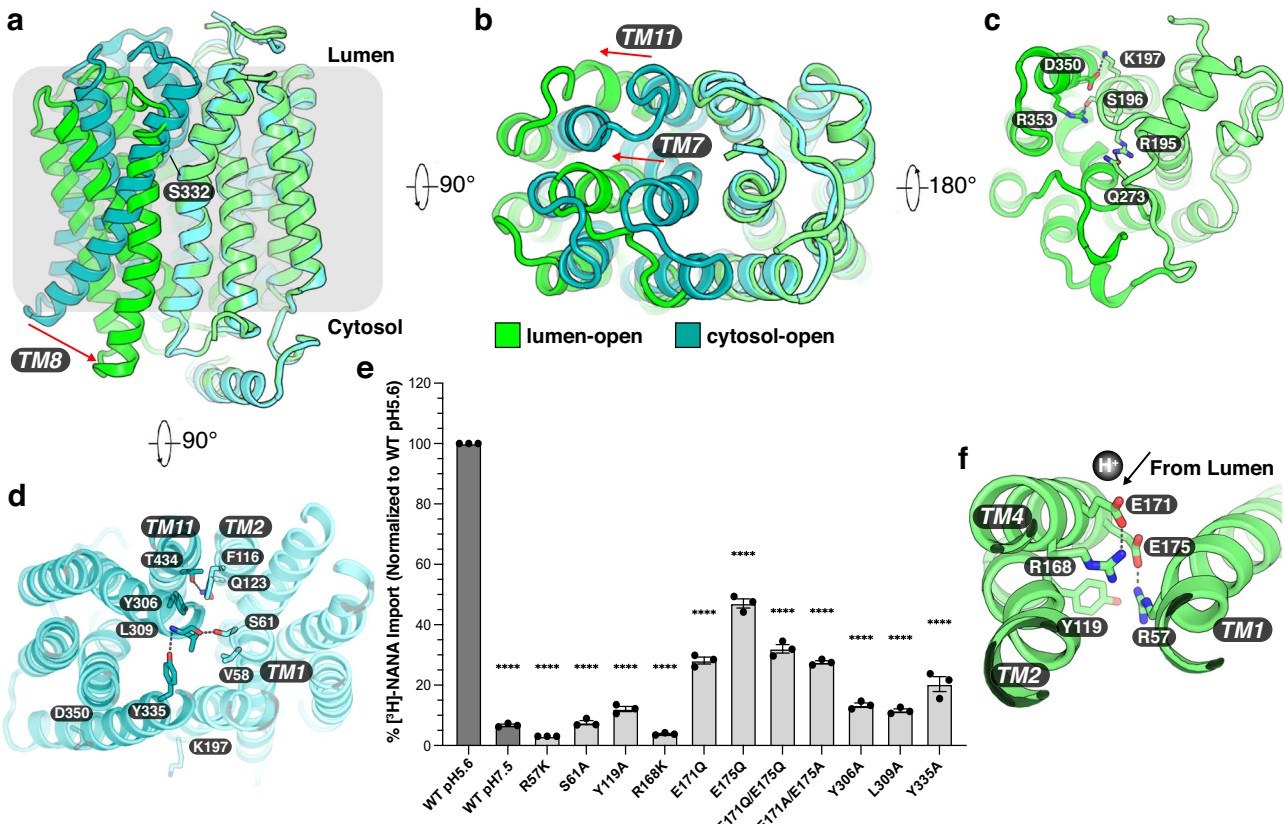

**Fig. 2 | Transition of human Sialin in the transport cycle. a** Overall structural comparison between the cytosol-open state (cyan) and the lumen-open state (green). The arrow indicates the different arrangement of TM8. TMs and related residues are labeled. **b** The extracellular view. The arrow indicates the different arrangement of TM7 and TM11. **c** The cytosolic view of the lumen-open state. **d** The extracellular view of the cytosol-open state. The hydrophilic interactions between the structural elements are indicated by dashed lines. **e** Transport activity of mutants in key residues involved in structural transitions and proton sensing at pH 5.6. The wild type is defined as WT. Data are represented as mean ± SEM ($n = 3$ biological replicates). *P* values are all in comparison to WT pH5.6 and are all 3.0e-15 in order. An ordinary one-way ANOVA with Dunnett's multiple comparisons test was used to generate the statistics in GraphPad Prism 10. ****$P \leq 0.0001$. **f** Potential residues involving in proton coupling. E171 and E175 may sense a proton (black ball) and mediate the proton transfer.

concentration of 5 mM. The structure of NAAG-bound Sialin^WT was determined at 3.4-Å resolution (Fig. 5a, Supplementary Fig. 8, and Table 1). An elongated density was found within the positively charged cavity that opens into the cytosol that was absent from the apo structures (Fig. 5b). The NAAG fits the density well, so we built the NAAG according to the morphology of this density (Fig. 5a). Tyr54, Gln207, Ser411 and His414 form hydrophilic interactions with NAAG (Fig. 5c and Supplementary Fig. 8e).

To measure Sialin-mediated NAAG and glutamate transport we used an assay similar to previous studies on glutamate transport by VGLUT2[30]. We incubated tritium labeled NAAG (Fig. 5d) with cells expressing Sialin^L/G in Ringer's solution for 30 min. The cells were then washed to remove excess NAAG and were incubated with Ringer's solution for 1 h to allow for NAAG efflux. We also incubated tritium-labeled glutamate with cells expressing Sialin^L/G in Ringer's solution for 15 min. The cells were then washed to remove excess glutamate and were incubated with Ringer's solution for 5 min to allow for glutamate efflux. Our results showed that the cells with the overexpression of Sialin^L/G transported NAAG and glutamate out of the cells at both pH 5.6 and pH 7.5 (Fig. 5d, e). Notably, the transport of both substrates was inhibited by Fmoc-Leu-OH at a final concentration of 100 μM (Fig. 5d, e).

Our transport assay shows that the alanine substitution on Tyr54, Gln207 or His414 causes over 50-90% reduction in NAAG transport (Fig. 5f). In comparison to the apo cytosol-open structure, the conformation of Phe179 on TM4 changes in order to create a cavity that is suitable for accommodating the acetyl-Asp of NAAG (Fig. 5g). In the apo lumen-open state, Phe179 on TM4 is in a similar position to that in the apo cytosol-open state, and the conformation change of C-domain induces a shift of TM10 towards the center of Sialin causing the collapse of NAAG-binding pocket (Fig. 5h). Compared to the Fmoc-Leu-OH-bound state, Phe179 presents a notable conformational change to create the aromatic interaction network with other residues for engaging Fmoc-Leu-OH; Gln207 shifts away from the cavity, avoiding steric hindrance with the inhibitor (Fig. 4d).

Moreover, there is a putative density around Tyr54, Tyr119, and the side chain of NAAG-acetyl-Asp (Supplementary Fig. 9a–c). A cation might be modeled into this density and may form polar interactions with Tyr119 and Thr178 and the carboxy group of acetyl aspartate of NAAG (Supplementary Fig. 9c). To validate the function of this cation in the binding site, we conducted MD simulations. Intriguingly, when the cation was modeled as potassium, the interaction energy between NAAG and nearby residues remained around ~−20 kcal/mol over a time scale of 100 ns. However, in the presence of a sodium ion or in the absence of a cation, the interaction energy decreased to 0 within a time scale of 50 ns (Supplementary Fig. 9e, f and Movies 2, 3). The MD simulations revealed that at the 50 ns mark, NAAG had moved out of the binding site when there was no ion or with a sodium ion (Supplementary Fig. 9e, f and Movies 2, 3). On the other hand, NAAG was able to remain confined within the binding site in the presence of a potassium ion (Supplementary Fig. 9g and Movie 4), implying that potassium may facilitate the engagement of NAAG in the cytosol. However, this hypothesis should be further validated by other approaches.

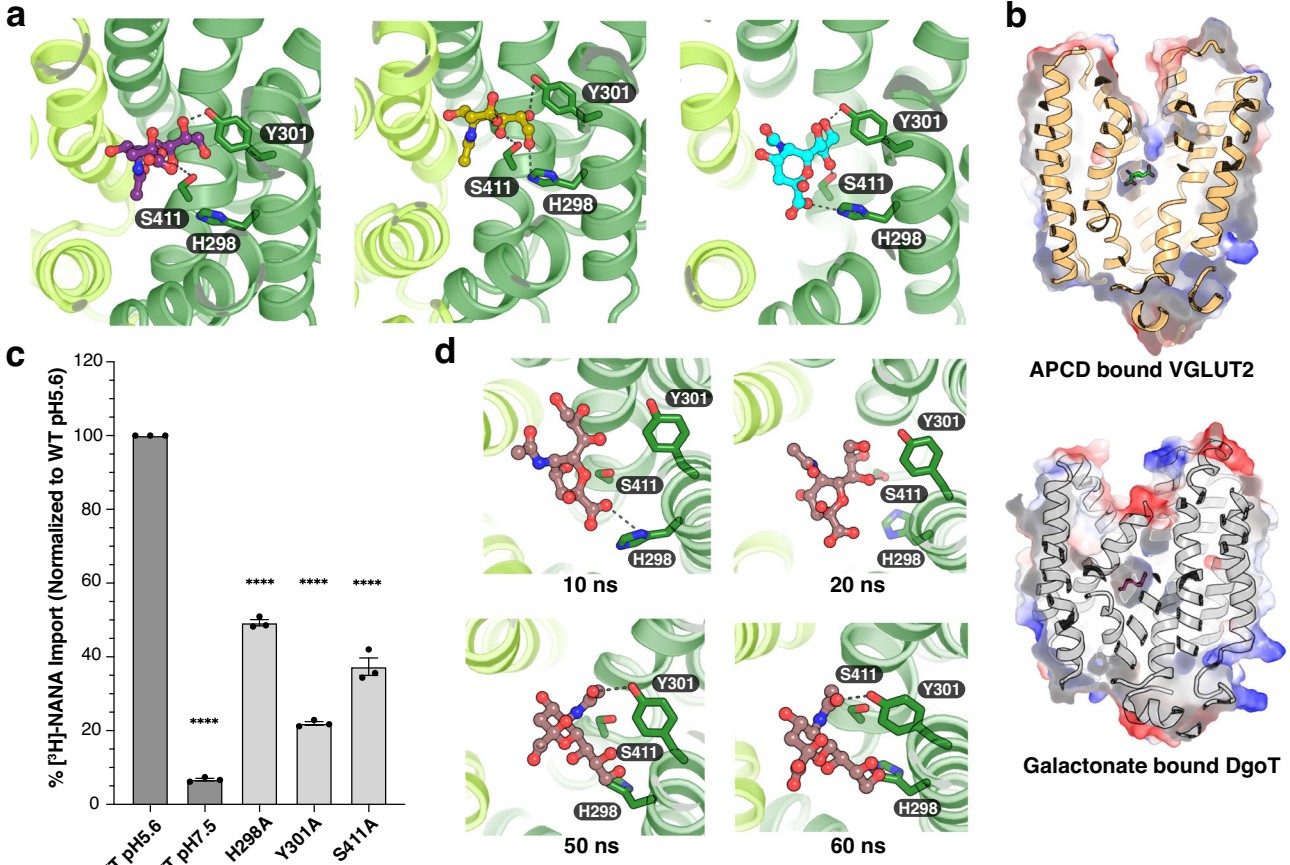

**Fig. 3 | A putative sialic acid binding site in the lumen-open Sialin. a** The top three docking result of sialic acid-bound Sialin[R168K] from Glide. Interactions between sialic acid and residues in the lumen-open cavity. The sialic acid ligand is shown as sticks. **b** Surface representation of L-trans-1-amino-1,3-dicarboxy cyclopentane (ACPD)-bound VGLUT2 (PDB-7T3N, top) and galactonate bound DgoT (PDB-6E9O bottom). The ligands are shown as sticks. **c** Transport activity of mutants in key residues potentially involved in NANA engagement at pH 5.6. The wild type is defined as WT. Data are represented as mean ± SEM ($n$ = 3 biological replicates). $P$ values are all in comparison to WT pH 5.6 and are 5.9e-12, 5.1e-8, 9.0e-11, and 2.4e-9 in order. An ordinary one-way ANOVA with Dunnett's multiple comparisons test was used to generate the statistics in GraphPad Prism 10. ****$P \le 0.0001$. **d** Snapshots of interaction details between NANA and Sialin residues during 100 ns MD simulations at the indicated time points.

A previous study showed that Sialin also functions as an aspartate and glutamate transporter in hippocampal synaptic vesicles and pineal synaptic-like micro-vesicles[11]. We used NAAG-bound structure as a model to dock aspartate and glutamate into Sialin in the presence of sodium using Glide[28] (Supplementary Fig. 10a, b). Owing to the small size of aspartate and glutamate, our docked models suggest that Gln207 may be not involved in the engagement of aspartate and glutamate. Our functional analysis showed that Sialin[Q207A] is able to transport glutamate but not NAAG supporting this model (Fig. 5e, f). Based on the structural and sequence similarity, it is tempting to speculate that the other SLC17 transporters bind their neurotransmitter substrates in a similar fashion (see discussion).

## Discussion

Many lysosomal transporters, like cystinosin[31], SLC7A14[32], and Spns1[33], transport their substrates using the proton gradient. A previous study showed that Asp46 and Glu133 of DgoT (equivalent to Leu56 and Glu175 of Sialin) play vital roles in pH sensitivity[34]. We speculate that Glu171 and Glu175 may serve as proton sensors and mediate proton coupling, similar to the role of Glu133 in DgoT. To test this hypothesis, we mutated Glu171 and Glu175 as well as surrounding residues (Arg57, Tyr119, Arg168) that stabilize the conformations of these putative proton sensors. Although all the mutants correctly expressed on the cell surface (Supplementary Fig. 5b), Sialin[E171A/E175A] and Sialin[E171Q/175Q]

exhibited a reduction of over 60% in transport efficiency (Fig. 2e), which, along with the previous study on the pH sensing mechanism of Sialin[19], is consistent with the idea that these residues participate in proton coupling. However, the previous study, which used an insect cell-based transport assay, showed no transport activity for the Sialin[E171A/E175A] mutant[19]. A possible reason for this difference is that we used HEK cells in our transport assay. Our assays exhibited the existence of residual activity suggesting additional residues may be required. Mutations on Arg57, Ser61, Tyr119, or Arg168 severely impaired the function of Sialin (Fig. 2e). It is likely these residues not only participate in proton coupling but are also involved in the state transitions of Sialin. Indeed, the S61A and R168K mutations confine Sialin in the lumen-open state (Fig. 1d), indicating the importance of these residues in regulating the conformation of Sialin.

Through our structural and functional studies, we have established a molecular blueprint that enables a mechanistic interpretation of mutations that cause both Salla disease and ISSD. Since Arg39 forms a salt bridge with Glu262 and Lys136 binds to the carbonyl of Val248, two Salla disease-causing mutations (Arg39Cys and Lys136Glu) destabilize the cytosolic opening of Sialin (Supplementary Fig. 6a). The ISSD-causing mutations (Gly328Glu, Pro334Arg and Gly371Val) introduce steric hindrance between the transmembrane helices in the C-domain, possibly interfering with transporter dynamics (Supplementary Fig. 6b).

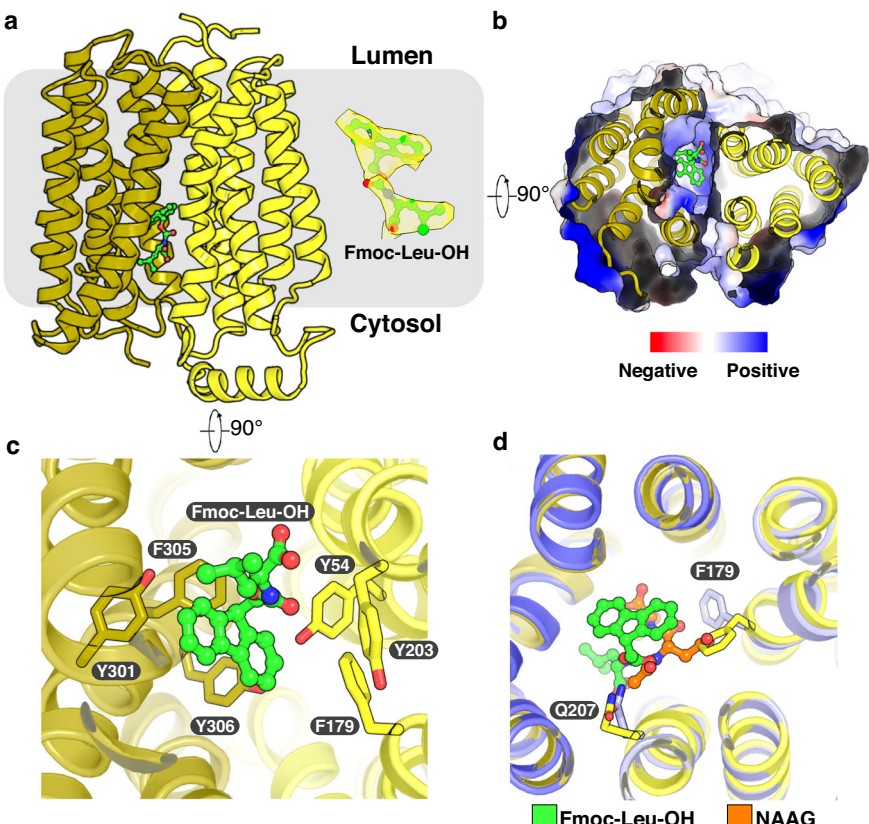

**Fig. 4 | Sialin structure in complex with Fmoc-Leu-OH. a** Overall structure of Fmoc-Leu-OH bound Sialin[WT] viewed from the side of the membrane. The cryo-EM map of Fmoc-Leu-OH is shown at threshold 0.375. **b** Surface representation of Sialin from the cytosolic side. **c** Interactions between Fmoc-Leu-OH and residues in the cytosol-open cavity. **d** The structural comparison between NAAG-bound Sialin and Fmoc-Leu-OH bound Sialin.

Glutamate serves as a primary excitatory neurotransmitter in the central nervous system of mammals and plays a critical role in fundamental processes such as learning, cognition, and memory. Although the structures of VGLUT2 have been reported recently[17], it remains unknown how glutamate is transported. Since Sialin and VGLUT2 are both members of the SLC17 family, our findings may provide a molecular proxy for the transport mechanism of VGLUT2. Since there are no notable conformational changes in the N-domain during the transport cycle, the conformation of the C-domain is likely a key factor in controlling the translocation pathway (Fig. 6). Our analyses show that Tyr135 of VGLUT2 adopts the same position as Tyr119 of Sialin and Tyr195 is in an analogous conformation to Phe179 in the apo structure of Sialin (Fig. 5e and Supplementary Fig. 10c). Tyr135 of VGLUT2 adopts the same position as Tyr119 of Sialin and Tyr195 is in an analogous conformation to Phe179 in the apo structure of Sialin. Notably, with the exception of Glu171 in Sialin (equal to Gln187 in VGLUT2), the key residues involved in conformational transition in Sialin are conserved in VGLUT2 (Supplementary Fig. 10d), implying that the SLC17 transport family may adopt a similar fashion to transport neurotransmitters. Nevertheless, further investigation into the SLC17-mediated neurotransmitter transport is required to validate this hypothesis.

The measurements of Sialin transport in this work were conducted using cell-based assays. NAAG, which accumulated within the cells, might be metabolized into N-acetylaspartic acid and glutamate during the 90-minute incubation in our experiments. However, this concern can be mitigated by the results of mutating Gln207, a residue that is essential for NAAG engagement but does not affect glutamate transport. Indeed, this mutation abolished Sialin-mediated NAAG transport (Fig. 5d–f), indicating that in those particular experiments, we measured NAAG flux but not glutamate (coming from degraded NAAG) flux. In vitro transport assays, such as proteoliposome experiments, may further validate our findings. We tentatively modeled a potassium ion in the NAAG-bound structure and our MD simulations provide insights into the potential role of this cation. Using proteoliposome assays to precisely control the ion environment would be beneficial to clarify the function of this cation in NAAG transport.

## Methods

### Sialin transport assays
Human Sialin with an N-terminal GFP tag and surface localizing mutations L22G/L23G was obtained from C. Anne. The Human TRPML1 (L[15]L and L[577]L to alanine residues, termed as TRPML1[L/A]), which expresses on the surface, was subcloned to pEGFP-C1 as a negative control for transport assays[35]. HEK293 cells were seeded onto poly-D-lysine (Gibco) coated black polymer-bottomed 24-well plates (Ibidi) at a density of 80 k cells per well in DMEM low glucose (Sigma) with 5% FCS (Gemini Bio) and 1% penicillin-streptomycin (Gibco). 24 h later, the cells were transfected with the Sialin[L/G] constructs using Fugene6 (Promega). 48 h after the transfection the cells were used for the radiolabeled transport assays.

Cells were used a total of 48 hours post-transfection. The medium was vacuumed off and the EGFP fluorescence of each well was measured using a Synergy Neo2 (BioTek). The fluorescence was measured in an area scan format, and the average of all the reads per well was used as the final fluorescence measurement for normalization.

For the radiolabeled sialic acid uptake assay, the cells were then gently rinsed with 500 μl of high pH buffer containing 20 mM Tris HCl pH 7.5, 150 mM NaCl, 1 mM $MgSO_4$, and 5 mM glucose. This buffer was then removed, and the uptake reaction was initiated with the addition of either high pH buffer (see above), or low pH buffer (consisting of

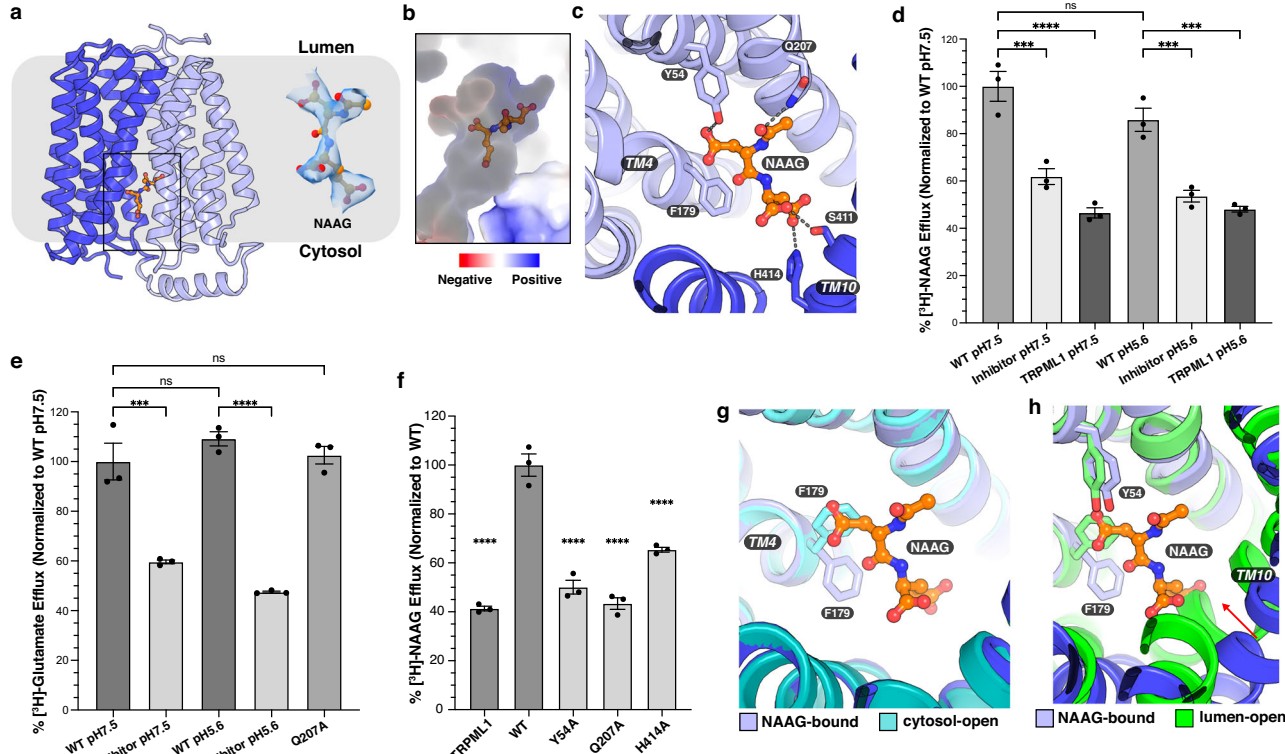

**Fig. 5 | Sialin structures in complex with NAAG. a** Overall structure of NAAG-bound Sialin[WT] viewed from the side of the membrane. The cryo-EM map of NAAG is shown at threshold 0.243. **b** The NAAG in the cytosol-open cavity. **c** Interactions between NAAG and residues in the cavity. **d** Functional validation of human Sialin[L/G] NAAG transport. Data are represented as mean ± SEM ($n = 3$ biological replicates). $P$ values are 0.000144, 4.58e-6, 0.164063, 0.000686, and 0.000158 in order. **e** Functional validation of human Sialin[L/G] glutamate transport. Data are represented as mean ± SEM ($n = 3$ biological replicates). $P$ values are 0.000189, 0.495929, 0.988875, and 4.32e-6 in order. For panels (**d**) and (**e**), inhibitor was added at 100 μM concentration and statistics were generated using an ordinary one-way ANOVA with Tukey's multiple comparisons test in GraphPad Prism 10. ns $P > 0.05$, ***$P \le 0.001$, ****$P < 0.0001$. **f** Transport activity of mutants in key residues involved in NAAG engagement at pH 7.5. Data are represented as mean ± SEM ($n = 3$ biological replicates). $P$ values are all in comparison to WT and are 1.0e-7, 4.6e-7, 1.4e-7, and 1.35e-5 in order. An ordinary one-way ANOVA with Dunnett's multiple comparisons test was used to generate the statistics in GraphPad Prism 10. ****$P \le 0.0001$. **g** Comparison of Phe179 in cytosol-open apo state and NAAG-bound states. **h** Comparison of NAAG-bound state and lumen-open apo state with the red arrow showing the movement of TM10.

20 mM MES pH 5.6, 150 mM NaCl, 1 mM MgSO₄, and 5 mM glucose), both containing 20 nM ³H-N-acetylneuraminic acid (American Radiolabeled Chemicals). The reactions were incubated at room temperature for 10 min. The reaction buffer was then removed, and the cells were washed two times with 500 μl each of their respective buffers without the radiolabeled substrate. The cells were then lysed with 200 μl RIPA buffer (Pierce) with a 1:1000 concentration of Benzonase Nuclease (Sigma-Aldrich). This lysate was then added to 5 ml of Complete Counting Cocktail (Research Products International) and the radioactivity was measured by liquid scintillation counting using a Tri-Carb 2800TR (PerkinElmer).

For the NAAG radiolabeled transport assay, the cells were incubated with high pH Ringer's solution (145 mM NaCl, 10 mM glucose, 10 mM HEPES pH 7.5, 4 mM KCl, 2 mM CaCl₂, 1 mM MgCl₂) containing 36 nM radiolabeled ³H-N-Acetyl-Aspartyl-Glutamate (American Radiolabeled Chemicals). The cells were incubated for 30 minutes at 37 °C in 8.8% CO₂ to allow for the uptake of NAAG into the cells. The hot buffer was then aspirated, and the cells were gently washed three times with 500 μl ice-cold high pH Ringer's solution. The cells were then incubated with 250 μl of pre-warmed Ringer's solution at either high pH or low pH for 1 h at 37 °C with 8.8% CO₂. The low pH Ringer's solution consisted of 145 mM NaCl, 10 mM glucose, 10 mM MES pH 5.6, 4 mM KCl, 2 mM CaCl₂, 1 mM MgCl₂. After one hour, the buffer was added to 5 ml of Complete Counting Cocktail (Research Products International) and the radioactivity was measured by liquid scintillation counting using a Tri-Carb 2800TR (PerkinElmer).

For the glutamate radiolabeled transport assay, the buffer conditions and steps were the same as the NAAG transport assay with the following differences. Cells were incubated with high pH Ringer's solution containing 42 nM radiolabeled ³H-Glutamic Acid (Revvity) for 15 min at 37 °C in 8.8% Co₂. After 15 min, they were then washed three times with ice-cold high pH Ringer's solution, and then incubated with 250 μl of pre-warmed high pH or low pH Ringer's solution for 5 min at 37 °C with 8.8% CO₂. The buffer was then measured by liquid scintillation counting using a Tri-Carb 2800TR (PerkinElmer).

### Sialin cellular localization
HEK293 cells were seeded into 24-well plates (Corning) at a density of 80k cells/well in DMEM low glucose (Sigma) with 5% FCS (Gemini Bio) and 1% penicillin-streptomycin (Gibco). The next day, cells were transfected with plasmids encoding N-terminal GFP tagged Sialin with L22G/L23G (Sialin[L/G]) and a plasmid encoding Spns2-mCherry wild-type[25] using and following the protocol for Fugene6. 48 hours after transfection, the cells were quickly trypsinized (Gibco), the digestion was stopped after 2 min, and the cells were applied to a glass-bottomed 8-well slide (Ibidi) and fluorescence images were acquired using a Zeiss LSM 800 microscope system (Zeiss).

### Protein expression and purification of human Sialin
The complementary DNA (cDNA) encoding either full-length human wild-type Sialin (Sialin[WT]), human Sialin with an R168K mutation (Sialin[R168K]), or human Sialin with an S61A mutation (Sialin[S61A]), all with

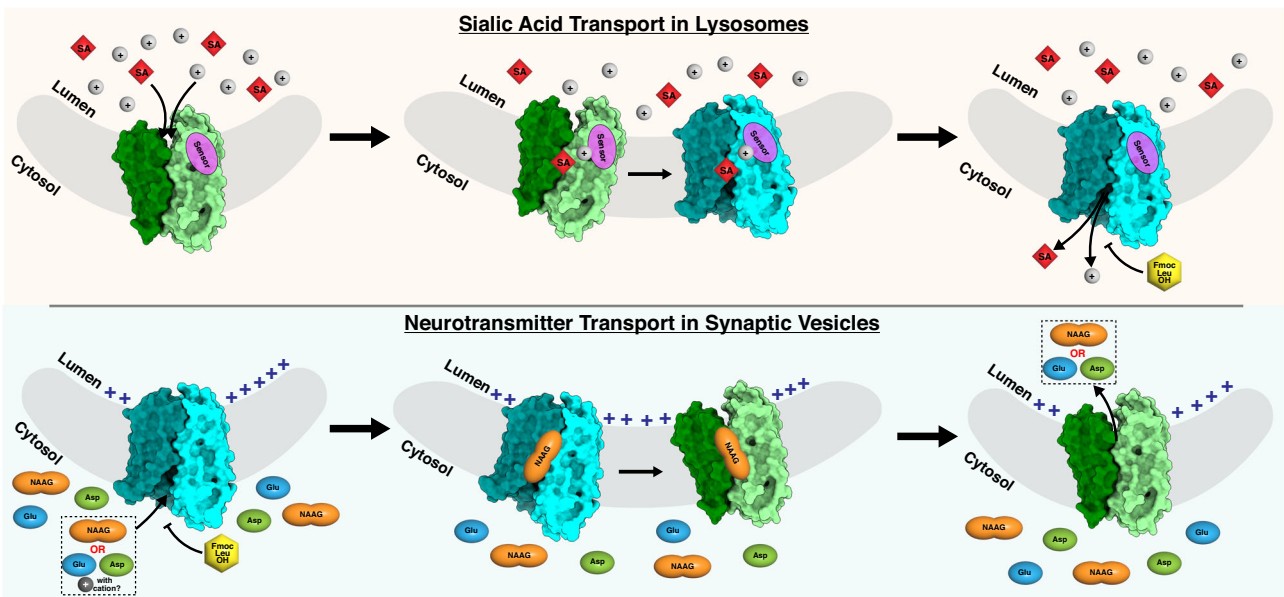

**Fig. 6 | The working models of Sialin.** In lysosomes, Sialin senses the proton in the lumen by its putative proton sensor, which is a cluster of residues in the N-terminal domain. The sialic acid (SA) binds to the lumen-open cavity. While the proton is translocated into the cytosol, Sialin changes to the cytosol-open state, and sialic acid is delivered into the cytosol. In synaptic vesicles, the neurotransmitter binds to the cytosol-open cavity. The membrane potential triggers the conformational change of Sialin to release the neurotransmitter into the lumen. The inhibitor Fmoc-Leu-OH binds to the cytosol-open cavity and confines the conformation of Sialin, preventing its transport activity.

an N-terminal FLAG tag, was cloned into the pEG BacMam vector[36] using primers outlined in Supplementary Table 2. The protein was expressed using baculovirus-mediated transduction of mammalian HEK-293S GnTI- cells (ATCC no. CRL-3022) grown at 37 °C for 48 h post-transduction. Protein samples used to obtain the apo cytosol-open Sialin[WT] pH 7.5, Fmoc-Leu-OH-bound, Sialin[R168K], and Sialin[S61A] structures used the following protocol. Cells were disrupted by sonication in lysis buffer (20 mM HEPES pH 7.5, 150 mM NaCl, 1 mM PMSF, and 5 mg/mL leupeptin). After a 10-min centrifugation at 3220 × g, the supernatant was incubated with 1% (w/v) Lauryl Maltose Neopentyl Glycol (LMNG, Anatrace) at 4 °C for 1 h. The lysate was clarified by a 30-minute centrifugation at 38,759 × g, and the resulting supernatant was loaded onto a Flag-M2 affinity column (Sigma-Aldrich). The resin was washed twice with wash buffer containing 20 mM HEPES pH 7.5, 150 mM NaCl, and 0.01% MNG. The protein was eluted by wash buffer with 100 mg/mL 3xFlag peptide. The elution was then concentrated and purified by size exclusion chromatography (SEC) on a Superdex 200 Increase column (Cytiva). The SEC buffer contained 20 mM HEPES pH 7.5, 150 mM NaCl, and 0.06% (w/v) Digitonin (ACROS Organics). The apo cytosol-open Sialin[WT] pH 5.0 was purified using the same lysis, wash, and elution buffer, but the SEC buffer contained 20 mM sodium acetate pH 5.0, 150 mM NaCl, and 0.06% (w/v) Digitonin. To prepare the protein sample used for the NAAG-bound structure, N-Acetyl-Asp-Glu (NAAG, Sigma) was made to a stock solution of 100 mM and added throughout the purification in each buffer to final contraction of 0.2 mM. The SEC buffer used contained 20 mM HEPES pH 7.5, 150 mM NaCl, 0.06% (w/v) Digitonin (ACROS Organics), and 0.2 mM NAAG. To assemble the Sialin·Fab[10B8] complex, purified Sialin was incubated with Fab[10B8] at a 1:1.2 molar ratio at 4 °C for 30 min. The mixture was then purified by SEC using the same buffers used for SEC of Sialin alone. Peak fractions were collected and concentrated for grid preparation.

## Antibody generation

IgG-10B8, a mouse monoclonal anti-human Sialin antibody, was prepared by fusion of SP2-mIL6 mouse myeloma cells (ATCC no. CRL-2016) with splenic B lymphocytes obtained from BALB/c mice (n = 1) at

UT Southwestern with the approval of the Institutional Animal Care and Research Advisory Committee #2017-102391 as previously described[37]. Briefly, mice were immunized with one primary and four boosts of purified recombinant human Sialin reconstituted with liposomes in phosphate buffered saline (PBS) combined with Sigma Adjuvant System. We used the combined techniques of ELISA, immunoblot analysis, and immunoprecipitation to identify antibodies that are preferentially bound to Sialin in its native but not SDS-denatured state. To clone IgG-10B8, total RNA was isolated from the hybridoma by RNA extraction kit (Qiagen) following the manufacturer's protocol. Total RNA was subjected to reverse transcription reactions using a Superscript III reverse transcription kit (Invitrogen), and the resultant cDNA was used as a template in PCR reactions with degenerate primers to amplify the variable regions. Sequences of the resulting PCR products were analyzed with the IMGT database (http://www.imgt.org/) to determine the variable regions of the light chain and heavy chain, which were then cloned into shuttle vectors for the light chain and the Fab region of the heavy chain with a C-terminal 6xHis tag, respectively[38]. The resulting constructs were co-transfected to HEK-293S GnTI- cells (ATCC) using PEI MAX transfection reagent (Polysciences) for expression at 37 °C. After 72 h, the medium was harvested and applied to Ni-NTA gravity columns. Following several washes with buffer A (20 mM HEPES, 150 mM NaCl, pH 7.5) containing 20 mM imidazole, bound material was eluted in buffer A containing 250 mM imidazole. The eluate was then applied to a Superdex-200 Increase size-exclusion chromatography column (GE Healthcare) in buffer A, and peak fractions containing Fab[10B8] were collected for complex assembly.

## Cryo-EM sample preparation and data acquisition

Protein samples were concentrated before grid preparation. For the NAAG-bound structure, the stock solution of NAAG was adjusted to pH 7.0 by NaOH and supplemented to the protein sample purified with NAAG throughout, to an additional final concentration of 5 mM. For the Fmoc-Leu-OH sample, stock solution was made to 50 mM in DMSO, and inhibitor was added to final concentration of 1 mM. 3 µl of each

protein sample was applied to R1.2/1.3 400 mesh Au holey carbon grids (Quantifoil). After 3 s, the grids were blotted for 4-5 s and plunged into liquid ethane using a Vitrobot Mark IV (FEI) operated at 22 °C and 100% humidity. The grids were loaded onto a 300 kV Titan Krios transmission electron microscope for data collection. Raw movie stacks for all structures except Sialin[S61A] were recorded with a K3 camera at a physical pixel size of 0.83 Å per pixel and a nominal defocus range of −0.8 to −2.2 µm. The exposure time for each micrograph was 5 s, dose-fractionated into 50 frames with a dose rate of ~1.2-1.4 e⁻/pixel/s (total dose ~60 electrons per Å²). Raw movie stacks for Sialin[S61A] were recorded with a Falcon 4i camera at a physical pixel size of 0.738 Å per pixel and a nominal defocus range of −0.8 to −2.2 µm. The exposure time for each micrograph was 4 s, with a total dose of ~60 electrons per Å².

### Cryo-EM image processing

The final collections consisted of: apo Sialin at pH 7.5 (1593 movies), apo Sialin at pH 5.0 (1242 movies), apo Sialin[R168K] at pH 7.5 (3206 movies), apo Sialin[S61A] at pH 7.5 (2925 movies), NAAG-bound cytosol-open (4014 movies), and Fmoc-Leu-OH bound cytosol-open (2 collections totaling 4356 movies). For each dataset except the apo Sialin[S61A], the dark subtracted images were first gain-normalized and corrected for beam-induced motion using MotionCor2 v1.2.1[39] performed in RELION 3.1[40]. The contrast transfer function (CTF) was estimated using CTFFIND4 v4.1.8[41]. Motion correction and CTF estimation for the apo Sialin[S61A] was performed using cryoSPARC Live v3.3.1. Auto picking was performed with crYOLO v1.7.6 (all datasets) and v1.9.4 (apo Sialin[S61A]) using the general model[42] with the particle threshold of 0.1-0.3 and a box size of 300 px. About 0.2–0.7 million particles were selected for each dataset. Subsequent 2D classification, multi-class ab initio reconstruction, and heterogenous 3D refinement were performed in cryoSPARC v3.3.0[43]. Particles of individual classes were then refined with nonuniform refinements[43], followed by local refinements with soft masks covering the Sialin-Fab[10B8] complex to further improve the map quality. The mask-corrected FSC curves were calculated in cryoSPARC, and reported resolutions are based on the 0.143 criterion. Local resolution estimations were performed in cryoSPARC.

### Model building and refinement

A predicted model of Sialin was generated by AlphaFold[44]. This model was then docked into the density map using ChimeraX v1.7[45]. The model was then refined iteratively using Coot v0.8.8[46], Phenix v1.17[47], and ISOLDE v1.7[48]. For the different conformations, the same model was used and manually fit using Coot, and iteratively refined using Phenix and ISOLDE. Structural model validation was performed using Phenix and MolProbity v4.3[49]. Figures were prepared using PyMOL v1.8.6.0 and ChimeraX. The docking was performed using Glide[28].

### Construction of model

The all-atom Sialin protein structure was modeled from the atomic coordinates of NAAG bound Sialin obtained at 3.4 Å resolution. Hydrogen atoms were added using H-build from CHARMM (47a2)[50]. Based on the structural data, one disulfide bond was constructed between Cys83 and Cys387. The pKa values of all titratable residues were determined by carrying out electrostatic energy computations with the protein pKa predictor PROPKA[51]. An initial protonation pattern was obtained to represent the protonation pattern at pH 7. NAAG (charge −3) and sialic acid (charge −1) were constructed in Avogadro (version 1.2) (https://avogadro.cc/); force field parameters were calculated using the program CGenFF program[52].

Four models were constructed: (1) Sialin-NAAG without a cation near NAAG, (2) Sialin-NAAG with Na⁺ near NAAG, (3) Sialin-NAAG with K⁺ near NAAG, (4) Sialin-NANA. For the fourth model, the structure was obtained by using Glide to dock NANA into the putative binding site (as described above). For each model, the protein structure was modeled in a lipid bilayer consisting of 100% 1-Palmitoyl-2-oleoyl-D-glycero-1-phosphatidylcholine (POPC), and the transmembrane domain (TMD) helices were inserted into the lipid bilayer using the CHARMM-GUI[53] and OPM database[54]. The protein-membrane system was next solvated with explicit TIP3 water[55] and K⁺ and Cl⁻ ions corresponding to 0.15 M KCl. Models 1-3 consisting of ~65,000 atoms was simulated in a rectangular box of dimension 75 Å x 75 Å x 123 Å; Model 4 consisting of ~114,000 atoms was simulated in a rectangular box of dimension 100 Å x 100 Å x 121 Å (Supplementary Table 3).

### Geometry optimizations and molecular dynamics

The initial geometry of each solvated protein-membrane complex was optimized using two rounds of 100 steps of steepest descent followed by 100 steps of adopted basis Newton-Raphson (ABNR) to remove any close contacts. All energy minimizations used the all-atom CHARMM36 parameters for proteins, lipids, and ions[56] and the TIP3P model for water molecules[55]. Next, the solvated protein-membrane complex was simulated with molecular dynamics (MD) at 310 K according to the following protocol with NAMD (3.0b4)[57]: 1) equilibration MD with Langevin dynamics (time step of 1 fs) for 375 ps followed by Langevin dynamics (time step 2 fs) for 1.5 ns; 2) production MD with Langevin dynamics (time step 2 fs) for 100 ns. To simulate a continuous system, periodic boundary conditions were applied. Electrostatic interactions were summed with the Particle Mesh Ewald method[58] (grid spacing ~0.93-0.96 Å; fftx 80, ffty 80, fftz 128). A non-bonded cutoff of 12.0 Å was used. Pressure was controlled using a Langevin piston (1.101325 bar), and temperature was controlled with Langevin dynamics with damping coefficient 1.0 ps⁻¹. Each simulation was performed once.

### Statistics and reproducibility

For data analysis, the background uptake and fluorescence were subtracted from the measured values. The uptake of each mutant was then normalized by their respective fluorescence values and then normalized to the Sialin[L/G] (sialic acid transport) or TRPML1[L/A] (NAAG transport) as 100%. A statistical analysis was calculated by ordinary one-way ANOVA with Dunnett's multiple comparisons test using GraphPad Prism 10 for all the functional assays except Figs. 5d−e which were calculated by ordinary one-way ANOVA with Tukey's multiple comparisons test using GraphPad Prism 10. All functional experiments were repeated at least three times on different days. Similar results were obtained. Quantification methods and tools used are described in each relevant section of the methods or figure legends.

### Reporting summary

Further information on research design is available in the Nature Portfolio Reporting Summary linked to this article.

## Data availability

The data that support this study are available from the corresponding authors upon request. The 3D cryo-EM density maps have been deposited in the Electron Microscopy Data Bank under the accession numbers EMD-41858 (apo Sialin at pH 7.5); EMD-41859 (apo Sialin at pH 5.0); EMD-41860 (apo Sialin[R168K]); EMD-41861 (NAAG-bound Sialin); EMD-41862 (Fmoc-Leu-OH-bound Sialin); and EMD-43984 (apo Sialin[S61A]). Atomic coordinates for the atomic model have been deposited in the Protein Data Bank under the accession numbers 8U3D (apo Sialin at pH 7.5); 8U3E (apo Sialin at pH 5.0); 8U3F (apo Sialin[R168K]); 8U3G (NAAG-bound Sialin); 8U3H (Fmoc-Leu-OH-bound Sialin); and 9AYB (apo Sialin[S61A]). Initial/final PDB states for MD simulations are provided as Source Data. The source data underlying Figs. 1g, 2e, 3c and 5d−f and Supplementary Fig. 1 are provided with this paper. Previously published structures referred to in this paper include: 8DWI, 6E9N, 4U4T, 8SBE, and 6LYY. Source data are provided with this paper.

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

## Acknowledgements

The Cryo-EM data were collected at the UT Southwestern Medical Center Cryo-EM Facility (funded in part by the CPRIT Core Facility Support Award RP170644). This research was supported in part by the computational resources provided by the BioHPC supercomputing facility located in the Lyda Hill Department of Bioinformatics, UT Southwestern Medical Center, TX. URL: https://portal.biohpc.swmed.edu. We thank C. Anne for providing the Sialin$^{L/G}$ plasmid, F. Liu for guiding the docking, Y. Qin and L. Esparza for tissue culture, and R. Wang and Y. Sun for their technical help and advice. This work was supported by NIH F31 HD110229 (to P.S.), NIH P01 HL160487, R35 GM149533, the Welch Foundation (I-1957), and the G. Harold and Leila Y. Mathers Foundation (MF-2302-03702) (to X.L.).

## Author contributions

P.S. performed the biochemical and structural experiments and conducted the transport assays. P.S. and L.D. screened the antibody. N.E.-M. performed the molecular dynamics simulations. P.S., N.E.-M., C.-H.L., and X.L. analyzed the data and contributed to the manuscript preparation. P.S. and X.L. conceived the research and wrote the manuscript.

## Competing interests

The authors declare no competing interests.
