## [Peer Review File · Nature Communications]

Structure and Inhibition of the Human Lysosomal Transporter SialinReviewer #1 (Remarks to the Author):

The lysosomal transporter for sialic acids, sialin, is defective in a rare leukodystrophy, Salla disease, and in early lethal infantile sialic acid storage disorders. Sialin has also been implicated in the transport of acidic hexoses, nitrate, and the excitatory neurotransmitters glutamate, aspartate and N-acetylaspartylglutamate (NAAG) in acidic organelles.

In this study, the authors characterized the cryo-EM structure of human sialin in two conformations, cytosol-open and lumen-open, in apo state and the structures of NAAG- and inhibitor-bound sialin in cytosol-open conformation. Combined with mutagenesis studies and sialic acid docking to the lumen-open cavity, they provide important insights into the structural basis for the polyspecificity of sialin and the distinct transport cycles underlying these activities. Their study also reveals a potential role for K⁺ in the binding, and possibly the transport, of NAAG and it provides structural evidence for the mechanism of non-competitive inhibitors with potential therapeutic interest.

Overall, this is a very interesting and well-executed paper supported by high-quality structural data. However, I have some issues regarding the NAAG transport assay. The comparison with prior mutagenesis studies and alternative interpretations of some findings also need clarification.

Major points:

1. It is uncertain whether the radioactivity efflux measured in the whole-cell [3H]NAAG efflux assay (equivalent to NAAG influx into synaptic vesicles) actually reflects NAAG transport because [3H]NAAG accumulated into the cells could be metabolized during the long incubation (overall 1h30) at 37°C. The authors should provide evidence that such metabolization is negligible in their experimental conditions. Otherwise, the radioactivity efflux may reflect the export of glutamate (which bears the tritium at its beta and gamma Carbons according to the manufacturer) if NAAG is cleaved by cytosolic peptidases within the cells, the export of a glutamate catabolite, or even a mixture of several export reactions.
2. A transport assay with the purified protein reconstituted in proteoliposomes would avoid such concern.
3. After establishing the reliability of the NAAG transport assay, it would be useful to compare the velocities of NAAG transport (or glutamate transport) and sialic acid transport to determine whether the two transport modalities operate at similar speeds.
4. The use of the R168K mutation to trap sialin in lumen-open state and the mechanism by which it weakens the luminal gate (Fig. 1c) are interesting. According to this model, a S161A mutation should directly disrupt the luminal gate. Did the authors characterize the S161A mutant by cryoEM to confirm the identity of the luminal gate?
5. In a previous cryo-EM study (ref. 19), the R168A mutant was ~60% active for sialic acid transport, in contrast with the lack of activity of sialin R168K in this study. Therefore, the introduction of a Lysine at position 168 might have a gain-of-function effect in addition to the loss of the Arg168 side chain/Arg57 carbonyl interaction shown in Fig. 1c. Does the Lys168 side chain interact with other residues or backbone groups in the lumen-open structure? This aspect should be discussed. A luminal view of sialin R168K similar to Fig. 1c would also help.
6. The two studies also substantially differ on the residual activity of the E171Q and E175Q mutants, and the E171Q/E175Q double mutant. In ref. 19, these mutations disrupted sialic acid transport whereas their effect is partial in Fig. 5e. This discrepancy should be discussed (the

sentence in lines 268-271 is misleading in this respect). While these residues may be involved in proton sensing, the substantial residual activity observed by the authors challenges a role for Glu171 and Glu175 in 'mediating proton coupling' (lines 264-265).

7. The putative role for K⁺ in NAAG binding (Suppl. Fig. 7) is intriguing. The alternative interpretations provided (K⁺ symport, or an allosteric role in sialic acid binding without co-transport; lines 175-177) should be easily discriminated with a proteoliposome transport assay.

8. The predominance of H298, Y301 and S411 in coordinating sialic acid (Fig. 4a) would be strengthened by providing the frequency of these interactions during molecular dynamics of the sialin-sialic acid complex.

9. It is unclear which working hypothesis the authors favor for the mechanism of NAAG, glutamate or aspartate transport. In figure 6, the schematic depicts a uniport mechanism driven by the membrane potential of the synaptic vesicle, whereas the main text suggests that proton coupling (H⁺/NAAG antiport) is needed to complete the transport cycle (lines 282-283). The authors should clarify this ambiguity.

Minor points:

1. The position of the labeling in the [3H]NAAG molecule (see above) should be provided in the manuscript.

2. Abstract, lines 18-19: '... but also transport mono and *diacidic* neurotransmitters' rather than 'diamine'

3. Introduction, lines 68-70: '... mechanism for *importing* neurotransmitters [...] from the cytosol *into* synaptic vesicles' rather than 'exporting'

4. Fig. 1a: the view of the WT pH 5.0 structure from the cytosol mentioned in the legend is missing in panel a.

5. Results, line 249, '... seal the substrate translocation pathway from the cytosolic leaflet': from the cytosolic 'space' or 'compartment' seems more appropriate for hydrophilic substrates binding to/from an aqueous compartment.

Reviewer #2 (Remarks to the Author):

Comments on "Structural Insights into the Dual-Transport Ability of Sialin" by Schmiede et al

The manuscript by Schmiede, et al. reports the structure determination by single-particle cryo-EM of distinct functional states of the human transporter Sialin, which belongs to the SLC17 subfamily of transporters. Sialin has been reported to be able to transport both sialic acid from the lysosome to the cytoplasm, and to export diverse neurotransmitters from the cytosol to synaptic vesicles. Mutations in this transporter are responsible for two neurodegenerative sialic acid storage disorders. Thus, a mechanistic understanding of the function of this protein is highly relevant. The authors used a monoclonal Fab fragment specific for Sialin, making possible the determination of high-resolution structures of the protein in distinct conformational states, including a structure with inhibitor bound. The manuscript is well written and the structural data and analysis are of high-quality and well presented.

In principle, I would approve publication of the manuscript, after the authors address some major and minor concerns listed below:

Major concerns:

1. I would call the attention to re-examine the uniqueness of Sialin as a protein that uses a pH-driven mechanism for sialic acid transport, and a membrane potential-driven mechanism for NAAG transport, since the data indicating that there is no pH-driven NAAG transport is weak. I would not be surprised if proton co-transport would occur during the export of NAAG since the same residues that are protonated-deprotonated during sialic acid uptake are exposed to two environments with different pH during NAAG transport (cytoplasm=high-pH / synaptic vesicle lumen=low-pH). Furthermore, the single point mutations of residues E171, R168, and E175, affect strongly both, the import of sialic acid and the export of NAAG. I strongly recommend the authors to perform proton co-transport assays in proteoliposomes reconstituted with purified Sialin, which could give a stronger support to the absence of proton-transport during NAAG export.
2. Please describe if the transport assays were performed with a mixture of hot/cold substrates. In the methods section, it is written as if only 20nM NAAN or 36nM NAAG were used. This is critical, since (i) when loading NAAG into cells (presumably through Sialin), the concentration gradient would not be high-enough to allow transport against the membrane potential; (ii) the concentration of inhibitor used for the assays is in a 1:100 – 1:300 molar excess over substrate. This makes difficult the interpretation about whether this is a good inhibitor or not.
3. Is Fmoc-Leu-OH not transported? is the inhibition 'competitive' without net transport of this molecule? If so, why?
4. Lines 159-173. Based on MD-simulations, the authors suggest that the density observed close to the carboxy group of the acetyl aspartate of NAAG is a K⁺ ion. Have the authors tested a hydronium ion? this same site allocates proton-binding residues in other MFS transporters. Please explain.
5. The strong effect of the mutant R168K is striking since both residues are positively charged and the interaction with the carbonyl of the backbone would still be possible. Can the author propose an alternative explanation? Perhaps, this residue is part of the proton transport relay. A comparison of the location of this residue and the suggested proton-binding residues in this and other characterized MFS transporters might shed some light on this point.
6. Lines 169-173. Have the authors tested the effect of different concentration of potassium on NAAG transport?

Minor concerns:

- Structures of Sialin with bound NAAG and the inhibitor Fmoc-Leu-OH show density for both of these molecules but not density of the surrounding residues. Please show. Also, please indicate the threshold used in Chimera to generate this figures.
- The density close to NAAG was assigned to a cation. Are there other residues contacting this potential cation? Please show the surrounding residues.
- Line 405- ...final concentration...
- Fig.3 'd' should be changed to 'c'
- Fig 4 'd' to be changed to 'c'

Reviewer #1 (Remarks to the Author):

The lysosomal transporter for sialic acids, sialin, is defective in a rare leukodystrophy, Salla disease, and in early lethal infantile sialic acid storage disorders. Sialin has also been implicated in the transport of acidic hexoses, nitrate, and the excitatory neurotransmitters glutamate, aspartate and N-acetylaspartylglutamate (NAAG) in acidic organelles.

In this study, the authors characterized the cryo-EM structure of human sialin in two conformations, cytosol-open and lumen-open, in apo state and the structures of NAAG- and inhibitor-bound sialin in cytosol-open conformation. Combined with mutagenesis studies and sialic acid docking to the lumen-open cavity, they provide important insights into the structural basis for the polyspecificity of sialin and the distinct transport cycles underlying these activities. Their study also reveals a potential role for K⁺ in the binding, and possibly the transport, of NAAG and it provides structural evidence for the mechanism of non-competitive inhibitors with potential therapeutic interest.

Overall, this is a very interesting and well-executed paper supported by high-quality structural data. However, I have some issues regarding the NAAG transport assay. The comparison with prior mutagenesis studies and alternative interpretations of some findings also need clarification.

Major points:

1. It is uncertain whether the radioactivity efflux measured in the whole-cell [³H]NAAG efflux assay (equivalent to NAAG influx into synaptic vesicles) actually reflects NAAG transport because [³H]NAAG accumulated into the cells could be metabolized during the long incubation (overall 1h30) at 37°C. The authors should provide evidence that such metabolization is negligible in their experimental conditions. Otherwise, the radioactivity efflux may reflect the export of glutamate (which bears the tritium at its beta and gamma Carbons according to the manufacturer) if NAAG is cleaved by cytosolic peptidases within the cells, the export of a glutamate catabolite, or even a mixture of several export reactions.

Response: We agree with this reviewer that the long incubation may introduce degradation of NAAG. Although we have dedicated tremendous efforts in troubleshooting the proteoliposome assay (please see our response in Point 2), it appears that this assay still does not consistently yield reliable results in our hands. We have expanded our discussion in the "Limitations of this study" section to emphasize that future investigations incorporating the proteoliposome assay would be beneficial in providing additional support for our model.

To validate our structural observations, in addition to the ³H-NAAG efflux assay, we also performed the cell-based ³H-glutamate efflux assay. We showed that Sialin^{WT} can transport glutamate in the cell-based assay, as well as Sialin^{Q207A}. Notably, Sialin^{Q207A} cannot transport NAAG out of cells, indicating that residue Q207 is required for NAAG transport (Fig. 5e and f) but not for glutamate. This is in full agreement with our structural observation and docking prediction (Fig. 5c vs. Fig. S10a). Moreover, the fact that Sialin^{Q207A} reduces the efflux signal in the ³H-NAAG efflux assay suggests that in our assay, the signal likely reflects intact NAAG but not its degradation product (glutamate).

2. A transport assay with the purified protein reconstituted in proteoliposomes would avoid such

concern.

Response: We completely agree with this reviewer. A transport assay using proteoliposomes would directly address such concern. However, we have tried multiple approaches to reconstitute the protein into liposomes and perform the transport assay. Unfortunately, none of these attempts worked. Please see the summary of our assays below. As this reviewer can see, we attempted various assays with different buffer conditions, liposome reconstitution methods, and proteoliposome isolation methods; however, the assays seemed unstable, and the reproducibility was very poor. Therefore, we performed further cell-based transport assays to validate our observation instead of the proteoliposomes assay.

Liposome assay summary

a Liposome assay was using buffers to mimic the pH differential across the membrane necessary for NANA transport. This was done in order to validate our Liposome reconstitution methods as well as general assay protocol. Here, high pH (pH7.5) should be negative while low pH (pH5.0) should give positive results. **b** Assays used buffers intended to mimic the membrane potential differential across the membranes. The addition of Valinomycin should disrupt this membrane potential differential and thus lead to a negative result. *[Note: many of these assays have varying numbers of replicates due to trial-and-error testing of protocols and materials. This is just a representation of all the assays attempted.]*

3. After establishing the reliability of the NAAG transport assay, it would be useful to compare the velocities of NAAG transport (or glutamate transport) and sialic acid transport to determine whether the two transport modalities operate at similar speeds.

Response: Since we could not establish the NAAG transport assay, we are not able to address this concern. But previous studies (PMID: 21781115 and 23889254) suggest it took ~5 minutes for Sialin to transport either glutamate or sialic acid in their *in vitro* transport assay.

4. The use of the R168K mutation to trap sialin in lumen-open state and the mechanism by which it weakens the luminal gate (Fig. 1c) are interesting. According to this model, a S161A mutation should directly disrupt the luminal gate. Did the authors characterize the S161A mutant by cryoEM to confirm the identity of the luminal gate?

Response: That is a great idea. Since residue 161 is a valine, we think the reviewer was referring to serine 61. We have determined the structure of Sialin^{S61A} (Fig. 1e). Interestingly, the resulting structure reveals Sialin in a luminal-open conformation, which supports our hypothesis on the interaction network of luminal gate.

5. In a previous cryo-EM study (ref. 19), the R168A mutant was ~60% active for sialic acid transport, in contrast with the lack of activity of sialin R168K in this study. Therefore, the introduction of a Lysine at position 168 might have a gain-of-function effect in addition to the loss of the Arg168 side chain/Arg57 carbonyl interaction shown in Fig. 1c. Does the Lys168 side chain interact with other residues or backbone groups in the lumen-open structure? This aspect should be discussed. A luminal view of sialin R168K similar to Fig. 1c would also help.

Response: The assay in ref. 19 was an insect cell-based transport assay whereas ours is a HEK cell-based assay. This difference may explain the discrepancy observed in these results. In our structure of Sialin^{R168K}, the Lys168 forms a salt bridge with Glu171 (Fig. 1d) presumably to block the proton sensing, causing the loss-of-function. It is also consistent with the structural observation on Sialin^{S61A}. We have included this part into the main text (lines 107-116) and showed the luminal view of Sialin^{R168K} and Sialin^{S61A}, according to the suggestion of this reviewer.

6. The two studies also substantially differ on the residual activity of the E171Q and E175Q mutants, and the E171Q/E175Q double mutant. In ref. 19, these mutations disrupted sialic acid transport whereas their effect is partial in Fig. 5e. This discrepancy should be discussed (the sentence in lines 268-271 is misleading in this respect). While these residues may be involved in proton sensing, the substantial residual activity observed by the authors challenges a role for Glu171 and Glu175 in ‘mediating proton coupling’ (lines 264-265).

Response: As we pointed out in point 5, the assay in ref. 19 was an insect cell-based transport assay which is different from our HEK cell-based assay. To figure out this discrepancy, we have

generated a Sialin^{E171A/E175A} construct and performed the sialic acid transport assay (Fig. 2e and Fig. S5). The result showed that the transport activity of Sialin^{E171A/E175A} is lower than that of Sialin^{E171Q/E175Q}. It suggests that Glu171 and Glu175 may contribute to the proton coupling but it is possible that the other residues may be involved in the proton coupling as well. We have included this data (Fig. 2e) and discussion in the revision (lines 269-275).

7. The putative role for K⁺ in NAAG binding (Suppl. Fig. 7) is intriguing. The alternative interpretations provided (K⁺ symport, or an allosteric role in sialic acid binding without co-transport; lines 175-177) should be easily discriminated with a proteoliposome transport assay. Response: Since we could not get the proteoliposome assay work, and due to the difficulty of altering ion concentrations in a cell-based assay, the function of potassium in NAAG transport remains unclear. We have expanded our discussion regarding K⁺ in the section of “Limitation of this study”.

8. The predominance of H298, Y301 and S411 in coordinating sialic acid (Fig. 4a) would be strengthened by providing the frequency of these interactions during molecular dynamics of the sialin-sialic acid complex.

Response: Point accepted; we have performed the MD simulations according to this reviewer’s suggestion (Movie S1). The result showed that on a 100 ns timescale, sialic acid moves within a few nanoseconds to a position that is stabilized with the interactions with His298, Tyr301, and Ser411. This stability is seen in the interactions between sialic acid and these three residues during the simulations. We have included snapshots from this simulation in the revision (Fig. 3d).

9. It is unclear which working hypothesis the authors favor for the mechanism of NAAG, glutamate or aspartate transport. In figure 6, the schematic depicts a uniport mechanism driven by the membrane potential of the synaptic vesicle, whereas the main text suggests that proton coupling (H⁺/NAAG antiport) is needed to complete the transport cycle (lines 282-283). The authors should clarify this ambiguity.

Response: We are sorry for this confusion. We did not propose the H⁺/NAAG antiport mechanism. As this reviewer can see in Fig. 5d-e, pH does not affect either NAAG or glutamate transport. Since it needs further investigations on the mechanism of how Sialin senses the membrane potential, we have revised the main text and figure to clarify this ambiguity (Fig. 6).

Minor points:

1. The position of the labeling in the [³H]NAAG molecule (see above) should be provided in the manuscript.

Response: We have shown the positions of the tritium labeling in [3H]sialic acid, [3H]NAAG and [3H]glutamate in Fig. S5a.

2. Abstract, lines 18-19: ‘... but also transport mono and *diacidic* neurotransmitters’ rather than ‘diamine’

Response: Point accepted. We have changed it.

3. Introduction, lines 68-70: ‘... mechanism for *importing* neurotransmitters [...] from the

cytosol *into* synaptic vesicles’ rather than ‘exporting’

Response: Point accepted. We have changed it.

4. Fig. 1a: the view of the WT pH 5.0 structure from the cytosol mentioned in the legend is missing in panel a.

Response: Point accepted. We have removed this figure legend.

5. Results, line 249, ‘... seal the substrate translocation pathway from the cytosolic leaflet’: from the cytosolic ‘space’ or ‘compartment’ seems more appropriate for hydrophilic substrates binding to/from an aqueous compartment.

Response: Point accepted. We have changed “leaflet” to “space” (line 151).

The authors thank the reviewer for their constructive comments and time.

Reviewer #2 (Remarks to the Author):

The manuscript by Schmiege, et al. reports the structure determination by single-particle cryo-EM of distinct functional states of the human transporter Sialin, which belongs to the SLC17 subfamily of transporters. Sialin has been reported to be able to transport both sialic acid from the lysosome to the cytoplasm, and to export diverse neurotransmitters from the cytosol to synaptic vesicles. Mutations in this transporter are responsible for two neurodegenerative sialic acid storage disorders. Thus, a mechanistic understanding of the function of this protein is highly relevant. The authors used a monoclonal Fab fragment specific for Sialin, making possible the determination of high-resolution structures of the protein in distinct conformational states, including a structure with inhibitor bound. The manuscript is well written and the structural data and analysis are of high-quality and well presented. In principle, I would approve publication of the manuscript, after the authors address some major and minor concerns listed below:

Major concerns:

1. I would call the attention to re-examine the uniqueness of Sialin as a protein that uses a pH-driven mechanism for sialic acid transport, and a membrane potential-driven mechanism for NAAG transport, since the data indicating that there is no pH-driven NAAG transport is weak. I would not be surprised if proton co-transport would occur during the export of NAAG since the same residues that are protonated-deprotonated during sialic acid uptake are exposed to two environments with different pH during NAAG transport (cytoplasm=high-pH / synaptic vesicle lumen=low-pH). Furthermore, the single point mutations of residues E171, R168, and E175, affect strongly both, the import of sialic acid and the export of NAAG. I strongly recommend the authors to perform proton co-transport assays in proteoliposomes reconstituted with purified Sialin, which could give a stronger support to the absence of proton-transport during NAAG export.

Response: The uniqueness of Sialin in the transport of different cargos has been shown by the previous studies (PMID: 21781115, 22778404 and 23889254). We have tried multiple approaches to reconstitute the protein into liposomes and perform the transport assay. Unfortunately, none of these attempts worked. Please see the summary of our assays in our response to Reviewer #1, Point 2. Our cellular transport assay results show that neither glutamate nor NAAG transport is affected by different pHs (Fig. 5d-e), which is different from Sialin-mediated sialic acid transport (Fig. 1g). In this revision, we removed the discussion of that Sialin employs its proton sensors to facilitate NAAG transport since this part should be further investigated by different approaches, and the main focus of our work is the structures in distinct states.

2. Please describe if the transport assays were performed with a mixture of hot/cold substrates. In the methods section, it is written as if only 20nM NAAN or 36nM NAAG were used. This is critical, since (i) when loading NAAG into cells (presumably through Sialin), the concentration gradient would not be high-enough to allow transport against the membrane potential; (ii) the concentration of inhibitor used for the assays is in a 1:100 – 1:300 molar excess over substrate. This makes difficult the interpretation about whether this is a good inhibitor or not.

Response: That is an interesting point. In this revision, we performed the cell-based assay to validate the Sialin-mediated NAAG and glutamate transport. The transport assays in this paper were performed with just hot substrates. While the concentrations seem low, they are in line with similar assays in previous studies on Sialin-mediated transport (PMID: 32608236 and 36662855). We assume that the loading of NAAG or glutamate into cells may not depend on Sialin, since there

are many other transporters on the cell surface. The transport of either NAAG or glutamate could be abolished by Fmoc-Leu-OH (Fig. 5d-e). The inhibitor which we used in this study has been validated by a previous study (PMID: 32608236). It showed that IC₅₀ of this inhibitor is 24 μM and it serves as a non-competitive inhibitor of sialic acid, which is consistent with our structural observation (Figs. 2 and 3).

3. Is Fmoc-Leu-OH not transported? is the inhibition ‘competitive’ without net transport of this molecule? If so, why?

Response: This point has been addressed by the previous study (PMID: 32608236, Figure 6). This study showed that Fmoc-Leu-OH functions as a non-competitive inhibitor of sialic acid, which is consistent with our structural observation.

4. Lines 159-173. Based on MD-simulations, the authors suggest that the density observed close to the carboxy group of the acetyl aspartate of NAAG is a K⁺ ion. Have the authors tested a hydronium ion? this same site allocates proton-binding residues in other MFS transporters. Please explain.

Response: That is an interesting point. A hydronium ion is difficult to test, however, because it has a lifetime on the order of picoseconds (10⁻¹² s). Our MD simulations indicate that the modeled ion (K⁺) is stable throughout the duration of the 100 ns (10⁻⁹ s) simulation. Based on the observed stability of K⁺, it is unlikely that this ion can be attributed to the much shorter-lived hydronium species.

5. The strong effect of the mutant R168K is striking since both residues are positively charged and the interaction with the carbonyl of the backbone would still be possible. Can the author propose an alternative explanation? Perhaps, this residue is part of the proton transport relay. A comparison of the location of this residue and the suggested proton-binding residues in this and other characterized MFS transporters might shed some light on this point.

Response: That is a great point, please see our response on the Point 4 of Reviewer #1, and our structure of Sialin^{S61A} (Fig. 1e). We have compared the structure with LacY (oligosaccharide: H⁺ symporter) and FucP (L-fucose:H⁺ symporter). As this reviewer can see on the right figure, the proton sensors in three structures are not conserved and the R168 (K168 in Sialin^{R168K}) is not conserved with the corresponding residues in either FucP or LacY. It is suggested that the proton sensors in the MFS transporters varies, therefore we did not place it in the revision.

6. Lines 169-173. Have the authors tested the effect of different concentration of potassium on NAAG transport?

Response: Since we used the cell-based to measure the transport of NAAG and it is difficult to change the intracellular concentration of potassium, it is not feasible for us to test the different

potassium concentrations. We have expanded our discussion regarding this in the “Limitation of this study”.

Minor concerns:

- Structures of Sialin with bound NAAG and the inhibitor Fmoc-Leu-OH show density for both of these molecules but not density of the surrounding residues. Please show. Also, please indicate the threshold used in Chimera to generate this figures.

Response: Point accepted. We have indicated the threshold for the NAAG and Fmoc-Leu-OH densities for Figs. 4 and 5 in the figure legends. We have also included figure panels in Figs. S7c and S8d that show the density of the surrounding residues. Thresholds for these figures are also in the figure legends.

- The density close to NAAG was assigned to a cation. Are there other residues contacting this potential cation? Please show the surrounding residues.

Response: Point accepted. We have shown Tyr119 and Thr178, which potentially contact this cation, in the Fig. S9c (line 235).

- Line 405- ...final concentration...

Response: Point accepted. We have changed it.

- Fig.3 ‘d’ should be changed to ‘c’

Response: Point accepted. We have changed it.

- Fig 4 ‘d’ to be changed to ‘c’

Response: Point accepted. We have changed it.

The authors thank the reviewer for their constructive comments and time.

Reviewer #1 (Remarks to the Author):

The authors satisfactorily addressed all concerns. The structure of the S61A is a great addition, confirming the interactions underlying the luminal gate. And the Q207A mutation is a good piece of evidence to discriminate between actual NAAG efflux and glutamate efflux (following NAAG metabolization within the cells) in the cell-based transport assay.

This study represents a significant contribution to the understanding of sialin transport mechanism – congratulation to the authors.

p. 6, line 114: there is a typo in 'salt bridge'

Reviewer #2 (Remarks to the Author):

The authors have addressed my previous concerns and adequately incorporated the new data into their analysis. Based on this, I recommend the publication of this manuscript.